# Body Mass Index and Overall Survival of Patients with Newly Diagnosed Multiple Myeloma

**DOI:** 10.3390/cancers14215331

**Published:** 2022-10-29

**Authors:** Bei Wang, Benjamin A. Derman, Spencer S. Langerman, Julie Johnson, Wei Zhang, Andrzej Jakubowiak, Brian C.-H. Chiu

**Affiliations:** 1Department of Public Health Sciences, University of Chicago, Chicago, IL 60637, USA; 2Section of Hematology/Oncology, Department of Medicine, University of Chicago, Chicago, IL 60637, USA; 3Center for Research Informatics, University of Chicago, Chicago, IL 60615, USA; 4Department of Preventive Medicine, Northwestern University Feinberg School of Medicine, Chicago, IL 60611, USA

**Keywords:** multiple myeloma, overall survival, obesity, body mass index

## Abstract

**Simple Summary:**

There is a growing need to clarify the effect of modifiable lifestyle factors such as obesity on outcomes of multiple myeloma (MM). In this paper, we examined the associations between body mass index (BMI) at different periods of life up to the time of diagnosis and overall survival among patients newly diagnosed with multiple myeloma. The key findings are that BMIs before and at the time of diagnosis were not associated with overall survival in MM, except that a higher BMI at diagnosis was associated with a better overall survival for females, irrespective of race/ethnicity. This is the first evidence that the BMI-survival association may differ by sex.

**Abstract:**

Obesity is associated with survival in several solid tumors and non-Hodgkin lymphoma, but its impact on multiple myeloma (MM) survival is unclear. We examined the associations between body mass index (BMI) at different periods of life up to the time of diagnosis and overall survival (OS) among 563 patients newly diagnosed with MM in 2010–2019. BMI at diagnosis was calculated using measured height and weight from electronic medical records (EMR). BMIs at age 20, maximum during adulthood, and 5 years before diagnosis were calculated using self-reported weights and measured height from EMR. Over a median follow-up of 49.3 months, 191 (33.93%) deaths were identified. We used multivariable Cox proportional-hazards models to examine the associations between BMIs and OS. Height as well as BMI before and at diagnosis was not associated with OS, but there is a U-shape association between weight and OS. Higher BMIs at diagnosis were associated with better OS among females (HR = 0.39 [0.22–0.71]), irrespective of race. In conclusion, our results suggest that BMI at different periods of life up to the time of diagnosis may not be associated with OS in MM, except that a higher BMI at diagnosis was associated with superior OS for females.

## 1. Introduction

Multiple myeloma (MM), a malignancy characterized by clonal expansion of malignant plasma cells in the bone marrow, affects more than 35,000 patients each year in the United States, and approximately 12,600 patients will die of the disease in 2022 [1,2]. With the introduction of novel treatment strategies, the 5-year relative survival of MM has increased to 58% [1]. Well-established prognostic factors include age, tumor burden (e.g., beta-2 microglobulin and albumin), high-risk cytogenetic abnormalities (HRCA), and elevated serum lactate dehydrogenase (LDH) [3,4]. In contrast, modifiable lifestyle factors such as obesity that could impact treatment outcomes of MM are understudied.

Obesity is associated with overall survival (OS) in several solid tumors [5,6] (e.g., breast, kidney, and lung) and non-Hodgkin lymphoma [7,8], but its associations with outcomes of MM is largely unknown. Being overweight and obese at the time of MM diagnosis was linked to lower mortality in a study of 2968 U.S. Veterans [9]. In contrast, two studies found no association between OS and being overweight or obese at the time of diagnosis or autologous stem cell transplantation [10,11]. One of the limitations of the previous studies is that most have limited data on well-established clinical prognostic factors. In addition, although obesity has been recognized as a risk factor for MM [12], no study has evaluated the association between pre-diagnostic obesity at different periods of life and the outcomes of MM. Recent studies have shown that excess body weight in early adulthood is associated with worse survival in breast cancer [13,14].

To clarify the association between obesity and overall survival of MM, we examined body mass index (BMI) at different periods of life up to the time of diagnosis and OS in a cohort of patients newly diagnosed with MM at the University of Chicago Medicine between 2010–2019.

## 2. Methods

### 2.1. Study Population

A total of 586 patients aged at least 18 and newly diagnosed with MM, according to the International Myeloma Working Group updated criteria [15], were prospectively enrolled from June 2010 through December 2019 at the University of Chicago Medicine. Eligible subjects were identified weekly from electronic admission medical records. We obtained written informed consent from each participant. The study was approved by the Institutional Review Board at the University of Chicago.

Participants completed a baseline self-administered questionnaire and donated research blood samples at the enrollment. The questionnaire collects information on pre-diagnostic height and weight at age 20, five years before diagnosis, and maximum adult weight, in addition to demographics, occupational history, lifestyle and health behaviors, dietary supplements, physical activities, personal and family medical history, and a semi-quantitative food frequency questionnaire. Height and weight measured at the time of diagnosis were obtained from electronic medical records (EMR). In addition, we collected baseline clinical, laboratory, and treatment data such as stage, monoclonal spike (M-Spike), estimated glomerular filtration rate (eGFR) [16], serum free light chains, LDH levels, among other additional pertinent laboratory results from EMR. Fluorescent in situ hybridization (FISH) analysis of CD138+ plasma cells enriched from bone marrow aspirate samples with a panel of probes targeting MM-associated cytogenetic abnormalities was performed at the University of Chicago Cancer Cytogenetic Laboratory.

### 2.2. Statistical Analyses

The primary goal of the study was to evaluate BMI at different periods of life up to the time of diagnosis and OS in MM. Vital status was ascertained using the National Death Index. OS was defined as the time from diagnosis until death from any cause. Participants were followed up from the date of diagnosis to the date of death, loss-to-follow, or administrative end date of 28 February 2022, whichever occurred first. The median follow-up length was 49.3 months.

BMI, an indirect measure of adiposity, was calculated as weight in kilogram (kg) divided by the square of height in meters (m^2^) [weight (kg)/height (m)^2^] [17], using weights during three different periods of a subject’s life and at the time of diagnosis. We used height measured at diagnosis from EMR in all BMI calculations because it is more objective than self-reported heights. Nevertheless, self-reported heights from the questionnaire were highly correlated with measured heights from the EMR (Pearson’s correlation coefficients: 0.92–0.95). We defined individuals as underweight (BMI < 18.5), normal weight (BMI 18.5–24.9, reference), overweight (BMI 25–29.9), and obese (BMI ≥ 30.0) based on the World Health Organization (WHO) guidelines [18]. Twenty-three subjects who were underweight at any time point(s) were excluded in all analyses due to the small group size. We also categorized BMI into quartiles based on its distribution among the at-risk cohort. The results were similar, and only the analyses using the WHO definition are presented. BMI, weight, and height are anthropometric factors that have been shown to be associated with health outcomes differently. It has been reported that BMI and body weight are different in their associations with pharmacokinetic factors, potentially leading to different responses to cancer treatment [19]. Therefore, we additionally modeled height and weights in quartiles based on their distribution in the at-risk cohort to examine the associations of height and weight with OS.

We compared characteristics of patients by vital status using mean (standard deviation) and proportions. The associations between height, weight, and BMI with OS were evaluated using hazard ratios (HRs) and 95% confidence intervals (CIs) from Cox proportional-hazards models. We first used age-adjusted univariate models to examine the association between each independent variable and OS. In multivariable analyses, the basic models included a priori adjustment for age (32–59/60–69/≥70; years), sex (female/male), and race/ethnicity (black/white) in the log-linear model component. The final models additionally adjusted for well-established prognostic factors, including ISS stage (1/2/3), LDH (normal <240/elevated: ≥240; U/L), and eGFR (≥60/<60; mL/min). M-spike, kappa/lambda ratio, and HRCA were not included in the final models because they were not associated with OS in the age-adjusted univariate analyses.

Previous findings suggest MM survival differs by sex and race/ethnicity [20,21,22]. Therefore, we conducted stratified analyses to evaluate sex- and race/ethnicity-specific BMI-OS associations. Potential heterogeneity of effect in the HRs for BMI on OS by sex or race/ethnicity was assessed by a Wald test of the cross-product terms. We used the log-rank for trend to examine the presence of linear trends of survivor function across the ordered groups of normal weight, overweight, and obese [23]. We also categorized BMI into quartiles based on sex-specific distribution among the at-risk females and males. The results were similar, and only the analyses using the WHO definition are presented. All statistical tests were two-sided, with *p*-values <0.05 considered statistically significant. Data analyses were performed using Stata (version 17.0) and R v4.1.2 R (Foundation for Statistical Computing, Vienna, Austria; https://cran.r-project.org, accessed on 1 September 2020).

## 3. Results

Over a median follow-up of 49.3 months (interquartile range: 23.5–87.0 months), a total of 191 (33.9%) deaths occurred in the patient cohort. Of the 563 subjects in this analysis, the mean age at diagnosis was 62.8 years old, 253 (44.9%) were female, and 147 (27.2%) were black. Compared with the at-risk participants, a larger proportion of the deceased participants were older and were Blacks. Clinical characteristics such as advanced ISS stage, lower eGFR, and higher LDH levels were associated with inferior survival (Table 1). The median survival for normal weight, overweight, and obese BMI status at diagnosis were 97.77, 119.26, and 103.07 months, respectively. The distributions of the clinical characteristics did not differ significantly by BMI status at diagnosis (Appendix A).

Table 2 provides the HRs and 95% CIs for all-cause mortality according to weight, height, and BMI at different life periods up to diagnosis. No statistically significant associations with OS were found for BMI at the time of diagnosis, BMI 5 years prior to diagnosis, and maximum BMI during adulthood. Results were similar between the basic and final models controlling for clinical prognostic factors. Interaction terms of BMI status at diagnosis and age (32–59/60–69/≥70; years), ISS stage (1/2/3), LDH (normal <240/elevated: ≥240; U/L), and eGFR (≥60/<60; mL/min) were tested in fully adjusted models and none of them was significant. Additionally, we found that height was not associated with OS. The second and third quartiles of weight at diagnosis and before diagnosis (i.e., weight five years before diagnosis and maximum weight in adulthood) were inversely associated with OS relative to the first quartile of weight. Significant weight changes (≥5%) from five years before diagnosis to the time of diagnosis was not associated with OS.

To examine whether the associations between BMI and OS differ by sex and race/ethnicity, we conducted stratified analyses (Table 3). The Kaplan–Meier survival curves (Appendix A) showed poorer OS in females and Blacks with lower BMIs. After adjusting for stage and other clinical prognostic factors (Table 3), in females, but not males, higher BMIs at the time of diagnosis were associated with a better OS (HR_overweight vs. normal weight_ = 0.64 [0.36–1.12]; HR_obese vs. normal weight_ = 0.39 [0.22–0.71]; *p*_trend_ =< 0.001; *p*_homogeneity_ = 0.01). There was also a suggestive trend of inverse association between a higher maximum BMI in a female’s adulthood and OS.

The suggestive inverse association between BMI at diagnosis and OS among Blacks remained in the multivariable models, although the test of homogeneity was not significant (HR_overweight vs. normal weight_ =0.64 [0.33–1.25]; HR_obese vs. normal weight_ =0.51 [0.26–1.00]; *p*_trend_ =0.05; *p_homogeneity_ =* 0.36) (Table 3). This may be due to a greater proportion of females in Blacks compared to the Whites in the current study (61.9% vs. 38.1%). Stratified analysis by both race/ethnicity and sex showed an inverse association between BMI at diagnosis and OS in both Black females (HR_obese vs. normal weight_ = 0.19 [0.08–0.46]; N = 86) and to a less extent in White females (HR_obese vs. normal weight_ = 0.44 [0.19–1.05]; N = 129), but not Black and White males (Appendix A).

## 4. Discussion

While we found little evidence that BMI at MM diagnosis or pre-diagnostic BMIs during different periods in life was associated with OS for the entire cohort, Blacks and females who had lower BMI had worse unadjusted OS compared to their counterparts. Multivariable analysis adjusting several known prognostic factors showed that higher BMI at diagnosis was associated with a better OS in females.

Findings from previous studies about BMI at diagnosis and OS of MM are not entirely consistent. In a study of 2968 U.S. Veterans (98% males), Beason et al. found that patients who were overweight and obese at the time of MM diagnosis had lower mortality compared with normal weight patients [9]. A recent study of patients with relapsed/refractory MM found that obesity was associated with a small OS benefit [24]. However, Jung et al. found no association between elevated BMI at MM diagnosis and OS [10]. Another study reported no overall effect of BMI at the time of autologous stem cell transplantation on OS and progression-free survival (PFS) but a better OS and PFS in obese patients receiving melphalan conditioning [11].

The inconsistencies might partially reflect differences in the study populations and methods. First, although the study populations are similar in terms of the age distributions, our cohort included newly diagnosed Black and White patients, whereas previous studies were limited to Veterans [9], Asians [10], or patients enrolled in clinical trials [11,24]. Second, our study has a longer follow-up duration compared to prior studies. Lastly, while all studies performed statistical adjustment for some clinical indices, data collection and inclusion of clinical indices are likely heterogeneous across studies. Nevertheless, adjustment for clinical prognostic factors had little impact on the associations between BMIs and OS in our study population. Taken together, it’s possible that obesity might be associated with higher risks of developing MM [12], but not necessarily more aggressive disease or outcomes once the disease has developed. Such paradox has been documented in several other cancers [25].

We found that higher BMIs at the time of diagnosis were associated with a better OS in females, but not males. The survival benefits limited to female patients may be related to a higher percentage of overall and peripheral adipose tissue in females than males [26], which may provide higher tolerance and energy reserve that may be important under the stress from intense anti-MM treatment [25,27]. It’s also possible that female patients with lower BMI at diagnosis may have experienced involuntary weight loss associated with MM in the years leading up to diagnosis. However, our results show that significant weight loss from five years before diagnosis to the time of diagnosis was not associated OS. In addition, females are much shorter than males in the current study. A shorter person with high adiposity may have a disproportionately higher BMI ratio than a tall person with high adiposity [28]. This may also explain why we observed a U-shaped association between weight and OS in the overall cohort. In addition, sex differences in circulating cytokines may also alter the impact of obesity. Adiponectin, an anti-inflammatory cytokine that has protective effects in MM [29,30], is higher in females [31]. Meanwhile, males have higher levels of interleukin-6 (IL-6) than females [32], which is a pro-inflammatory cytokine that inhibits apoptosis in myeloma cells [33]. However, people who are obese have lower levels of adiponectin and higher levels of IL-6 [34], which would be incongruent with our findings. Lastly, sex differences in pharmacokinetics may also play a role in the results. For example, a study has shown that docetaxel and doxorubicin dosed based on actual body weight resulted in increased drug exposure only in obese females but not males compared to their non-obese counterparts [35]. There is also evidence suggests that among patients with MM receiving high-dose melphalan, obese patients had similar body exposure to drug compared to non-obese patients despite receiving lower doses of melphalan [36]. With data suggesting that higher melphalan exposure is associated with superior outcomes, perhaps higher exposure among obese female patients contributes to the difference.

The present study has several strengths, including relatively long follow-up, comprehensive data on important clinical and prognostic factors from EMR, and prospective enrollment of newly diagnosed patients to minimize the potential of survival bias. There were also limitations to our study. We could not examine BMI after diagnosis and the impact of weight changes during the follow-up period, which might influence survival. Similar to other studies assessing BMI and MM, BMI is an indirect measure of adiposity, which plays an important role in the progression of MM. A study of newly diagnosed MM patients demonstrated that excessive visceral adipose measured by computed tomography correlates with poorer treatment response, advanced stage, and the presence of HRCA while BMI does not [37]. In addition, like other epidemiologic studies using EMR as sources of treatment data, we could not evaluate BMI with progression free survival because of difficulties in assessing response correctly after front-line treatment in medical records. Differences in therapeutic exposure in obese versus normal weight individuals might confound the observed association. Lastly, we could not validate self-reported weight at different life periods collected using the questionnaire. However, the self-reported height from the questionnaire and measured height from EMR were highly correlated, suggesting information bias in self-reported weight might be minimal.

## 5. Conclusions

Our results suggest that BMI at different periods of life up to the time of diagnosis may not be associated with OS in patients with newly diagnosed MM, with the exception that higher BMI was associated with superior OS for females. This novel finding requires confirmation. Randomized trials are needed to evaluate the effect of weight changes after diagnosis and during MM treatment on OS, MM-specific survival, and progression.

## Figures and Tables

**Table 1 cancers-14-05331-t001:** Demographics, clinical characteristics, age-adjusted univariate hazard ratios (HRs), and 95% confidence intervals (CIs) for all-cause mortality ^a^, UChicago Multiple Myeloma Epidemiology Study, 2010–2019.

Characteristic	Total	By Vital Status		
		At Risk N (%)	Death N (%)	HRs (95% CI)
		372 (66.07)	191 (33.93)	
Age at diagnosis (years)				
Mean (SD)	62.75 (10.11)	61.89 (10.04)	64.30 (10.19)	
<60	209 (37.12)	149 (40.05)	60 (31.41)	1.0 (Referent)
60–69	227 (40.32)	143 (38.44)	84 (43.98)	1.43 (1.02, 2.00)
≥ 70	127 (22.56)	80 (21.51)	47 (24.61)	1.83 (1.25, 2.69)
Sex				
Female	253 (44.94)	168 (45.16)	85 (44.50)	1.0 (Referent)
Male	310 (55.06)	204 (54.84)	106 (55.50)	1.10 (0.82, 1.46)
Race				
White	393 (72.78)	271 (77.21)	122 (64.55)	1.0 (Referent)
Black/African American	147 (27.22)	80 (22.79)	67 (35.45)	1.43 (1.06, 1.93)
Education				
Below college	102 (27.27)	60 (25.21)	42 (30.88)	1.0 (Referent)
Some college or completed college	175 (46.79)	118 (49.58)	57 (41.91)	0.81 (0.54, 1.21)
Graduate or professional degree	97 (25.94)	60 (25.21)	37 (27.21)	0.89 (0.57, 1.39)
International staging system (ISS)				
1	262 (64.06)	201 (70.28)	61 (49.59)	1.0 (Referent)
2	106 (25.92)	65 (22.73)	41 (33.33)	2.08 (1.40, 3.10)
3	41 (10.02)	20 (6.99)	21 (17.07)	4.51 (2.72, 7.48)
Estimated glomerular filtration rate (eGFR) (mL/min)				
≥ 60	365 (67.72)	255 (72.03)	110 (59.46)	1.0 (Referent)
<60	174 (32.28)	99 (27.97)	75 (40.54)	1.70 (1.26, 2.31)
Serum free light chains				
Low [<0.26]	116 (21.72)	68 (19.43)	48 (26.09)	1.82 (1.21, 2.75)
Normal [0.26–1.65]	144 (26.97)	100 (28.57)	44 (23.91)	1.0 (Referent)
High [>1.65]	274 (51.31)	183 (52.00)	92 (50.00)	1.35 (0.94, 1.95)
Elevated lactate dehydrogenase (LDH) levels (U/L)				
Normal (LDH < 240)	333 (73.51)	239 (77.35)	94 (65.28)	1.0 (Referent)
Elevated (LDH ≥ 240)	120 (26.49)	70 (22.65)	50 (34.72)	1.72 (1.22, 2.42)
Number of high-risk cytogenetic abnormalities ^b^				
0	116 (66.29)	77 (67.54)	39 (63.93)	1.0 (Referent)
1+	59 (33.71)	37 (32.46)	22 (36.07)	1.40 (0.83, 2.37)

SD, standard deviation. ^a^ HRs were calculated using the Cox proportional hazards regression models adjusting for age (<60, 60–69, ≥70). Numbers may not sum to total due to missing data. ^b^ High-risk MM-associated cytogenetic abnormalities include t(4;14), t(14;16), t(14;20), del(17/17p), and gain(1q) defined by the International Myeloma Working Group.

**Table 2 cancers-14-05331-t002:** Associations between height, weight, and body mass index (BMI) at multiple myeloma (MM) diagnosis and pre-diagnostic BMIs with overall survival among patients with MM.

		At-Risk (N)	Deaths (N)	Model 1 ^a^	Model 2 ^b^
				Hazard Ratio (95% confidence interval)	
At diagnosis					
				N = 508	N = 494
Height (m) ^c^	Q1 [1.35–1.63]	103	57	1.0 (Referent)	1.0 (Referent)
	Q2 [1.63–1.73]	98	57	0.99 (0.64, 1.52)	1.05 (0.67, 1.63)
	Q3 [1.70–1.80]	67	30	0.69 (0.37, 1.28)	0.90 (0.46, 1.74)
	Q4 [1.80–1.99]	90	42	0.82 (0.48, 1.20)	0.83 (0.43, 1.61)
	*p* Trend ^d^			0.42	0.40
Weight (kg) ^c^	Q1 [39.15–71.11]	87	60	1.0 (Referent)	1.0 (Referent)
	Q2 [71.11–81.91]	93	35	0.56 (0.34, 0.88)	0.59 (0.38, 0.93)
	Q3 [81.91–93.16]	88	39	0.60 (0.38, 0.94)	0.58 (0.36, 0.91)
	Q4 [93.16–194.85]	90	51	0.74 (0.47, 1.17)	0.83 (0.53, 1.32)
	*p* Trend ^d^			0.09	0.23
BMI	Normal	97	51	1.0 (Referent)	1.0 (Referent)
	Overweight	139	73	0.93 (0.64, 1.33)	0.89 (0.62, 1.29)
	Obese	122	62	0.87 (0.60, 1.28)	0.88 (0.60, 1.28)
	*p* Trend ^d^			0.23	0.37
5 years before diagnosis					
				N = 288	N = 280
Weight (kg) ^c^	Q1 [48.60–72.00]	47	29	1.0 (Referent)	1.0 (Referent)
	Q2 [72.00–84.15]	54	26	0.68 (0.38, 1.22)	0.62 (0.34, 1.13)
	Q3 [84.15–96.75]	49	18	0.56 (0.28, 1.10)	0.55 (0.27, 1.12)
	Q4 [96.75–171.00]	53	31	0.85 (0.45, 1.61)	0.86 (0.43, 1.70)
	*p* Trend ^d^			0.56	0.30
BMI	Normal	51	25	1.0 (Referent)	1.0 (Referent)
	Overweight	80	43	1.10 (0.67, 1.82)	1.10 (0.65, 1.87)
	Obese	66	36	1.16 (0.69, 1.95)	1.20 (0.69, 2.07)
	*p* Trend ^d^			0.65	0.74
Age 20					
				N = 280	N = 271
Weight (kg) ^c^	Q1 [42.75–58.50]	49	29	1.0 (Referent)	1.0 (Referent)
	Q2 [58.50–71.55]	50	26	0.75 (0.39, 1.50)	0.85 (0.40, 1.78)
	Q3 [71.55–81.90]	49	22	0.48 (0.20, 1.17)	0.51 (0.20, 1.34)
	Q4 [81.90–112.50]	50	24	0.68 (0.26, 1.76)	0.98 (0.34, 2.85)
	*p* Trend ^d^			0.38	0.23
BMI	Normal	130	61	1.0 (Referent)	1.0 (Referent)
	Overweight	51	30	1.14 (0.73, 1.80)	1.28 (0.80, 2.04)
	Obese	11	6	1.80 (0.75, 4.31)	2.65 (1.07, 6.54)
	*p* Trend ^d^			0.47	0.85
Adulthood maximum					
				N = 340	N = 329
Weight (kg) ^c^	Q1 [49.50–76.50]	54	35	1.0 (Referent)	1.0 (Referent)
	Q2 [76.50–90.00]	56	31	0.57 (0.33, 0.96)	0.49 (0.28, 0.86)
	Q3 [90.00–102.60]	65	30	0.57 (0.32, 1.01)	0.45 (0.25, 0.82)
	Q4 [102.60–225.00]	58	38	0.76 (0.43, 1.34)	0.78 (0.43, 1.39)
	*p* Trend ^d^			0.66	0.54
BMI	Normal	29	17	1.0 (Referent)	1.0 (Referent)
	Overweight	77	48	1.06 (0.60, 1.86)	0.93 (0.51, 1.68)
	Obese	118	69	0.88 (0.51, 1.51)	0.80 (0.45, 1.40)
	*p* Trend ^d^			0.28	0.24

HRs: hazard ratios; CI: confidence interval; BMI: body mass index; Q1–Q4: the 1st to the 4th quartiles. ^a^ Model 1: adjusted for age (<60, 60–69, ≥70), sex (male/female), and race (white/black); ^b^ Model 2: adjusted for age (<60, 60–69, ≥70), sex (male/female), race (white/black), international staging system (1/2/3), elevated lactate dehydrogenase levels (normal/elevated), and estimated glomerular filtration rate (≥60/<60). ^c^ The HRs of weight (kg) were calculated based on models that adjusts for height (m) measured at diagnosis from electronic medical records and other covariates in model 1 and model 2. ^d^ *p* Trend: *p* values of log-rank test for linear trend in survival function of patients of different BMI categories.

**Table 3 cancers-14-05331-t003:** Sex- and race- specific associations of body mass index (BMI) at multiple myeloma (MM) diagnosis and pre-diagnostic BMIs with overall survival among patients with MM in fully adjusted multivariable models ^a.^

	Female			Male			*p* Homo-Geneity ^c^	White			Black			*p* Homo-Geneity ^c^
	At-Risk (N)	Deaths (N)	HR 95% CI ^a^	At-Risk (N)	Deaths (N)	HR 95% CI ^a^		At-Risk (N)	Deaths (N)	HR 95% CI ^a^	At-Risk (N)	Deaths (N)	HR 95% CI ^a^	
At diagnosis														
			N = 215			N = 280				N = 354			N = 141	
Normal	47	31	1.0 (referent)	47	20	1.0 (referent)		74	34	1.0 (referent)	16	17	1.0 (referent)	
Overweight	44	26	0.64 (0.36, 1.12)	89	47	1.15 (0.67, 1.97)		97	49	0.99 (0.62, 1.57)	26	23	0.64 (0.33, 1.25)	
Obese	61	25	0.39 (0.22, 0.71)	58	37	1.57 (0.90, 2.74)		76	36	1.10 (0.68, 1.78)	36	25	0.51 (0.26, 1.00)	
*p* Trend ^b^			<0.001			0.06	0.01			0.89			0.05	0.36
5 years before diagnosis														
			N = 114			N = 167				N = 222			N = 59	
Normal	23	16	1.0 (referent)	28	9	1.0 (referent)		45	21	1.0 (referent)	6	4	1.0 (referent)	
Overweight	30	10	0.52 (0.21, 1.30)	44	33	2.01 (0.94, 4.32)		60	33	1.11 (0.61, 2.04)	14	9	0.59 (0.14, 2.54)	
Obese	27	13	0.74 (0.31, 1.79)	37	23	1.89 (0.85, 4.17)		47	23	1.16 (0.61, 2.19)	14	13	1.90 (0.54, 6.70)	
*p* Trend ^b^			0.16			0.54	0.06			0.94			0.32	0.97
Age 20														
			N = 107			N = 161				N = 216			N = 52	
Normal	61	29	1.0 (referent)	65	32	1.0 (referent)		104	48	1.0 (referent)	21	13	1.0 (referent)	
Overweight	14	5	0.92 (0.33, 2.58)	33	25	1.50 (0.85, 2.65)		37	24	1.17 (0.68, 1.99)	9	5	1.62 (0.50, 5.26)	
Obese	1	2	24.81 (4.41, 139.65)	10	4	1.72 (0.57, 5.17)		9	2	1.36 (0.32, 5.80)	1	4	9.25 (1.86, 46.08)	
*p* Trend ^b^			0.70			0.96	0.04			0.90			0.80	0.31
Adulthood maximum														
			N = 140			N = 190				N = 252			N = 78	
Normal	20	13	1.0 (referent)	9	4	1.0 (referent)		26	13	1.0 (referent)	2	4	1.0 (referent)	
Overweight	22	16	0.81 (0.36, 1.86)	51	32	1.45 (0.50, 4.20)		59	37	0.92 (0.46, 1.87)	12	11	0.55 (0.15, 2.06)	
Obese	52	25	0.45 (0.21, 0.97)	61	44	1.68 (0.59, 4.82)		83	43	0.83 (0.43, 1.63)	26	25	0.55 (0.18, 1.70)	
*p* Trend ^b^			0.03			0.57	0.08			0.43			0.28	0.76

HRs: hazard ratios; CI: confidence interval; BMI: body mass index; ^a^ HRs are based on models that include age (<60, 60–69, ≥70), sex (male/female), race (white/black), international staging system (1/2/3), elevated lactate dehydrogenase levels (normal/elevated), and estimated glomerular filtration rate (≥60/<60). ^b^ *p* Trend: *p* values of log-rank test for linear trend in survival function of patients of different BMI categories; ^c^ *p* Homogeneity: Wald tests of the cross-product terms of sex and race/ethnicity with BMI in fully adjusted multivariable models.

## Data Availability

The data can be shared up on request.

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
