# Peer review of "Body Mass Index and Overall Survival of Patients with Newly Diagnosed Multiple Myeloma"

_cancers, 2022, doi:10.3390/cancers14215331_

Round 1
Reviewer 1 Report
The paper reports on whether body mass index is related to overall survival in myeloma patients. Adult patients (aged 18y+, average age 62.7y) diagnosed with MM between June 2010 and December 2019 at the University of Chicago Medicine were followed via the US National Death Index until February 2022 (median follow-up 49.3 months; maximum follow-up of 11 years and 8 months). BMI was derived from weight and height measured at diagnosis, and also from self-reported weight at age 20, 5 years before diagnosis and maximum adult weight. The authors concluded that overall survival in myeloma patients varied by weight, being better at lower and higher weights and worse in between; and for women, OS was better among those who were obese compared to those of normal weight for their height.
The paper would benefit from some additional presentations. Please add K-M survival curves by BMI groups; consider providing supplementary material with the same BMI survival curves stratified by sex and by race. Report in the text the median survival time for the whole cohort, and in each BMI group. For Tables 1 to 3, add columns showing the OS (with 95% CI) for each subgroup of patients. Provide a table showing the distributions of patients by BMI group and the prognostic factors age, ISS, eGFR, serum free light chains, LDH and number of high-risk cytogenetic abnormalities.
HRs in Table 3 suggest that among females or blacks, OS is better among those who are obese compared to those who are normal weight; for males the opposite may be true while among whites there was no difference by BMI. With OS among females of normal weight being around 61% and 50% for blacks, it is these groups that have poorer survival than in the cohort overall (OS 66%) or compared to males (71%) or whites (70%) of normal weight. It would be worth considering- at least for stratification by sex- whether the association between BMI and OS among females is still present when using sex-specific quartiles for BMI (it is not clear from the methods whether sex-specific quartiles for all anthropometric measures were used; and whether those who were underweight were included in these calculations; please clarify).
The paper does not currently draw attention to the groups with poorer survival- why would women or blacks in the normal range of BMI fair worse not just than those of heavier weight but also compared to males and whites in the same BMI range? Groups with poorer prognosis should be highlighted in the abstract, results and discussion. Myeloma could be associated with weight loss in the lead up to diagnosis, - as well as during treatment- so those who are heavier at diagnosis may fair better. With information on weight at 5 years prior to diagnosis, is there the possibility of exploring changes in weight? Changes in weight in the lead up to diagnosis and the relevance to the findings for BMI at diagnosis should be discussed.
No clear rationale for exploring weight at different time periods before diagnosis is given. For five years prior to diagnosis, an explanation could be possible changes in weight in the lead up to diagnosis but this is not stated. Potential reasons to examine other weight variables are less transparent and it may be that weight more contemporaneous with diagnosis – or the lead up to- would have a more dominant effect on survival than weight at other timepoints. As such, the inclusion of weight variables at other timepoints/ages needs a strong rationale.
Height and weight were measured at diagnosis- was this available for all patients? Were there some patients who would be too poorly to be measured- if there were, please give numbers and the OS for the total cohort including these patients. Also, how does OS in this cohort compare with OS for MM in US national cancer statistics?
Please consider adding a totals column to Table 1 to describe the characteristics of the whole cohort.
Author Response
We appreciate the thoughtful comments.
- The paper would benefit from some additional presentations. Please add K-M survival curves by BMI groups; consider providing supplementary material with the same BMI survival curves stratified by sex and by race. Report in the text the median survival time for the whole cohort, and in each BMI group. For Tables 1 to 3, add columns showing the OS (with 95% CI) for each subgroup of patients. Provide a table showing the distributions of patients by BMI group and the prognostic factors age, ISS, eGFR, serum free light chains, LDH and number of high-risk cytogenetic abnormalities.
Response: We thank the reviewer for these important comments. We added a figure in the revision to present Kaplan-Meier survival curves by BMI status at the time of diagnosis (Figure 1). We also added supplementary figures for BMI survival curves stratified by sex and by race. As suggested by the reviewer, we added median survivals by BMI to the results section (line 168-169) to read: “The median survival for normal weight, overweight, and obese BMI status at diagnosis were 97.77, 119.26, and 103.07 months, respectively.” We have reported the OS with 95% CI in Tables 1 to 3 in the original submission. Finally, we added a Supplementary Table 1 (see below) as suggested by the reviewer showing the distributions of selected prognostic factors by BMI categories.
Supplementary Table 1. Selected prognostic factors of multiple myeloma by body mass index (BMI) status at the time of diagnosis in the UChicago Multiple Myeloma (MM) Epidemiology Study, 2010-2019
|
|
BMI status at diagnosisa |
P value of χ2b |
||
|
|
Normala |
Overweight |
Obese |
|
|
|
Number (%) |
|||
|
Age at diagnosis (years) |
||||
|
<60 |
56 (37.84) |
76 (35.85) |
71(38.59) |
.47 |
|
60-69 |
53 (35.81) |
91 (42.92) |
78 (42.39) |
|
|
>=70 |
39 (26.35) |
45 (21.23) |
35 (19.02) |
|
|
International staging system (ISS) |
||||
|
1 |
65 (61.90) |
104 (67.53) |
92 (63.89) |
.56 |
|
2 |
29 (27.62) |
40 (25.97) |
35 (24.31) |
|
|
3 |
11 (10.48) |
10 (6.49) |
17 (11.81) |
|
|
Estimated glomerular filtration rate (eGFR) (mL/min) |
||||
|
>=60 |
95 (67.86) |
140 (67.96) |
128 (70.33) |
.85 |
|
<60 |
45 (32.14) |
66 (32.04) |
54 (29.67) |
|
|
Serum free light chains |
||||
|
Low [<0.26] |
36 (25.90) |
43 (21.29) |
33 (18.23) |
.50 |
|
Normal [0.26-1.65] |
37 (26.62) |
51 (25.25) |
53 (29.28) |
|
|
High [>1.65] |
66 (47.48) |
108 (53.47) |
95 (52.49) |
|
|
Elevated lactate dehydrogenase (LDH) levels (U/L) |
||||
|
Normal [< 240] |
90 (78.95) |
131 (75.29) |
105 (67.31) |
.08 |
|
Elevated [>= 240] |
24 (21.05) |
43 (24.71) |
51 (32.69) |
|
|
Number of high-risk cytogenetic abnormalitiesc |
||||
|
0 |
29 (60.42) |
46 (65.71) |
40 (72.73) |
.41 |
|
1+ |
19 (39.58) |
24 (34.29) |
15 (27.27) |
|
|
a. BMI was calculated as weight in kilogram (kg) divided by the square of height in meters (m2) [weight (kg) / height (m)2], using weight and height measured at the time of diagnosis. BMI status was defined as normal weight (BMI 18.5-24.9, reference), overweight (BMI 25-29.9), and obese (BMI ≥30.0). b. P values for the Pearson’s chi-squared statistic c. High-risk MM-associated cytogenetic abnormalities include t(4;14), t(14;16), t(14;20), del(17/17p), and gain(1q) defined by the International Myeloma Working Group. |
||||
- HRs in Table 3 suggest that among females or blacks, OS is better among those who are obese compared to those who are normal weight; for males the opposite may be true while among whites there was no difference by BMI. With OS among females of normal weight being around 61% and 50% for blacks, it is these groups that have poorer survival than in the cohort overall (OS 66%) or compared to males (71%) or whites (70%) of normal weight. It would be worth considering- at least for stratification by sex- whether the association between BMI and OS among females is still present when using sex-specific quartiles for BMI (it is not clear from the methods whether sex-specific quartiles for all anthropometric measures were used; and whether those who were underweight were included in these calculations; please clarify).
Response: We ran additional analyses using sex-specific quartiles of BMI as suggested by the reviewer (see Table below). The results are consistent with those using quartiles based on the cohort overall. We clarified how quartiles were determined by revising the sentence in the method section (lines 145-147) to read “We also categorized BMI into quartiles based on sex-specific distribution among the at-risk females and males. The results were similar, and only the analyses using the WHO definition are presented”. Subjects who were underweight (n=23) at any time point(s) were excluded from the analysis due to small sample size. We have clarified this in the revision. Please see lines 119.
Table. Sex-specific associations of body mass index (BMI) at multiple myeloma (MM) diagnosis and pre-diagnostic BMIs with overall survival among patients with MM in fully adjusted multivariable modelsa
|
Female |
Male |
||
|
BMI quartilesb |
HR 95% CI |
BMI quartilesb |
HR 95% CI |
|
At diagnosis |
|
|
|
|
|
N=215 |
|
N=280 |
|
Q1 [18.8-24.2] |
1.0 (referent) |
Q1 [20.0-25.2] |
1.0 (referent) |
|
Q2 [24.2-27.7] |
0.64 (0.35, 1.19) |
Q2 [25.2-27.6] |
1.09 (0.60, 2.01) |
|
Q3 [27.7-32.7] |
0.29 (0.14, 0.58) |
Q3 [27.6-30.6] |
0.91 (0.51, 1.64) |
|
Q4 [32.7-61.5] |
0.52 (0.27, 1.00) |
Q4 [30.6-58.3] |
1.47 (0.85, 2.56) |
|
5 years before diagnosis |
|
|
|
|
|
N=114 |
|
N=167 |
|
Q1 [19.8-24.9] |
1.0 (referent) |
Q1 [19.6-25.5] |
1.0 (referent) |
|
Q2 [24.9-28.1] |
0.78 (0.29, 2.10) |
Q2 [25.2-28.0] |
1.84 (0.85, 3.99) |
|
Q3 [28.1-31.9] |
0.35 (0.12, 1.03) |
Q3 [28.0-31.0] |
1.36 (0.56, 3.33) |
|
Q4 [31.9-55.9] |
0.72 (0.27, 1.89) |
Q4 [31.0-51.1] |
1.91 (0.88, 4.13) |
|
Age 20 |
|
|
|
|
|
N=109 |
|
N=162 |
|
Q1 [18.1-20.0] |
1.0 (referent) |
Q1 [18.2-22.9] |
1.0 (referent) |
|
Q2 [20.0-22.2] |
0.59 (0.21, 1.67) |
Q2 [24.7-28.0] |
0.42 (0.16, 1.09) |
|
Q3 [22.2-24.7] |
0.95 (0.35, 2.58) |
Q3 [28.0-31.0] |
0.93 (0.46, 1.88) |
|
Q4 [24.7-43.5] |
1.26 (0.46, 3.44) |
Q4 [31.0-34.2] |
1.08 (0.53, 2.18) |
|
Adulthood maximum |
|
|
|
|
|
N=140 |
|
N=190 |
|
Q1 [19.3-26.4] |
1.0 (referent) |
Q1 [21.1-27.0] |
1.0 (referent) |
|
Q2 [26.4-30.9] |
1.05 (0.47, 2.33) |
Q2 [27.0-30.1] |
1.42 (0.68, 3.99) |
|
Q3 [30.9-35.6] |
0.42 (0.17, 1.01) |
Q3 [30.1-33.2] |
1.41 (0.69, 2.91) |
|
Q4 [35.6-62.2] |
0.70 (0.31, 1.61) |
Q4 [33.2-75.5] |
1.54 (0.76, 3.12) |
|
HRs: hazard ratios; CI: confidence interval; BMI: body mass index; Q1-Q4: the 1st to the 4th quartiles. a. HRs are based on models that include age (<60, 60-69, ≥70), sex (male/female), race (white/black), international staging system (1/2/3), elevated lactate dehydrogenase levels (normal/elevated), and estimated glomerular filtration rate (>=60/<60). b. Quartiles based on the distributions of BMI ratios of the at-risk population in females and males |
|||
- The paper does not currently draw attention to the groups with poorer survival- why would women or blacks in the normal range of BMI fair worse not just than those of heavier weight but also compared to males and whites in the same BMI range? Groups with poorer prognosis should be highlighted in the abstract, results and discussion. Myeloma could be associated with weight loss in the lead up to diagnosis, - as well as during treatment- so those who are heavier at diagnosis may fair better. With information on weight at 5 years prior to diagnosis, is there the possibility of exploring changes in weight? Changes in weight in the lead up to diagnosis and the relevance to the findings for BMI at diagnosis should be discussed.
Response:
We appreciated the reviewer’s valuable insights. We added the 3rd paragraph of the Results section (lines 194-195) to read: “The Kaplan-Meier survival curves (Supplementary Figure 1) showed poorer OS in females and Blacks who had lower BMIs. ” In results paragraph 4, we also added sentences (lines 200-203) to read: “The suggestive inverse association between BMI and OS among Blacks remained in the multivariable models, although the test of homogeneity was not significant (HRoverweight vs. normal weight =0.64 [0.33-1.25]; HRobese vs. normal weight =0.51 [0.26-1.00]; Ptrend=<.05; Phomogeneity=.36).
We ran additional analyses related to the weight change as suggested by the reviewer. We define weight changes from 5 years before diagnosis to diagnosis using the percentage of changes based on baseline weights, and then categorize weight changes into 1) within 5% change, 2) >=5% weight loss, and 3) >=5% weight gain. In total, there are 122 (40.53%) had within 5% change, 100 (33.22%) had weight loss >= 5%, and 79 (26.25%) had weight gain >= 5%. Overall, there was little evidence that weight changes were associated with OS (p=0.20) (see figure below). A comprehensive analysis of weight changes during treatment could be an interesting topic for future directions but is beyond the scope of the current paper. In results section paragraph 2 (line 191-192), we added “Significant weight changes (>=5%) from five years before diagnosis to the time of diagnosis was not associated with OS”.
We revised the 1st paragraph of the discussion (line 210-238) to read: “While we found little evidence that BMI at MM diagnosis or pre-diagnostic BMIs during different periods in life was associated with OS for the entire cohort, Blacks and females who had lower BMI had worse unadjusted OS compared to their counterparts. Multivariable analysis adjusting several known prognostic factors showed that higher BMI was associated with a better OS in females.” In addition, we added sentences in the Discussion section (line 270-273) explaining as why normal weight subjects may have poorer OS. “Reversely, female patients with lower BMI at diagnosis may have experienced involuntary weight loss associated with MM in the years leading up to diagnosis. Although our results show significant weight loss from five years before diagnosis to the time of diagnosis was not associated OS.”
- No clear rationale for exploring weight at different time periods before diagnosis is given. For five years prior to diagnosis, an explanation could be possible changes in weight in the lead up to diagnosis but this is not stated. Potential reasons to examine other weight variables are less transparent and it may be that weight more contemporaneous with diagnosis – or the lead up to- would have a more dominant effect on survival than weight at other timepoints. As such, the inclusion of weight variables at other timepoints/ages needs a strong rationale.
Response:
In addition to BMI, we evaluated weight at different time period up to the time of diagnosis because these anthropometric factors have been shown to associated with health outcomes differently. Although correlated with weight, BMI is an estimate of body fatness (i.e., a measure of adiposity). Furthermore, it has been reported that BMI and body weight are different in their associations with pharmacokinetic factors (Paulzen et al., 2016), potentially leading to different responses to cancer treatment. Recent studies (Charvat et al., 2022, Arnold et al., 2019) show early excess body weight is associated with worse survival in breast cancer. We have added this rationale in the Introduction (line 73-75). Also, in Methods section (lines 122-126), we added: “BMI, weight, and height are anthropometric factors that have been shown to be associated with health outcomes differently. It has been reported that BMI and body weight are different in their associations with pharmacokinetic factors (Paulsen et al., 2016), potentially leading to different responses to cancer treatment. Therefore, we additionally modeled height and weights in quartiles based on their distribution in the at-risk cohort to examine the associations of height and weight with OS.” We further addressed the potentials of weight loss leading to poorer OS in both results and discussion sections (see response to item 3).
- Height and weight were measured at diagnosis- was this available for all patients? Were there some patients who would be too poorly to be measured- if there were, please give numbers and the OS for the total cohort including these patients. Also, how does OS in this cohort compare with OS for MM in US national cancer statistics?
Response:
Our UChicago Multiple Myeloma Epidemiology Study enrolled only newly diagnosed patients with multiple myeloma seen at the University of Chicago. Height and weight were measured at their first or second clinical appointment. As such, we do not expect many missing data on height and weight. Indeed, among the 586 participants included in this analysis, measured height was missing for only 19 subjects and no patient had missing measured weight. The median survival of the overall cohort including these patients is 108.5 months (same as the sample excluding these patients). We compared OS in our cohort to OS for MM in the US. According to the SEER-Medicare database (2007-2014), the median survival for MM patients was 64.8 and 54 months for Black and White patients, respectively (Ailawadhi et al., 2019). The median survival was 108.5 months in our cohort. A better OS in our cohort is likely due to different era of coverage: our study was initiated later (2010) and as such, a greater proportion of patients have access to novel therapeutics. Also, patients tended to be treated with 3 or 4 combinations of regimens as their front-line treatment at the University of Chicago Medicine.
- Please consider adding a totals column to Table 1 to describe the characteristics of the whole cohort.
Response: we have added a total column in Table 1 as suggested by the reviewer.
Reviewer 2 Report
Wang et al describe a cohort of patients with MM treated at their institution and look at the association between obesity as measured by BMI and survival in patients with MM. They find no overall association between obesity at any time and MM survival. But, when looking at women and men separately they find an association between obesity and improved outcomes in women only. This is a fascinating finding, and if confirmed will require additional research on the biological underpinnings. The authors should be commended on their clear and well written paper that I think will be of interest to myeloma community at large. I have a few comments/questions that I believe would help improve the quality of the manuscript.
The biggest question for me is why there might be these differences in survival associated with obesity. Is this due to myeloma related outcomes? One way to look at this is whether lower BMI is associated with any adverse prognostic factors (HR cytogenetics, ISS stage, LDH, lower GFR). Does the normal BMI group include women who are borderline underweight, potentially with aggressive disease mediating weight loss that predisposes the referent population to worse survival?
Related to this, why is that overweight females live longer? Do they have better myeloma outcomes? Do they have better PFS1 or better ORR to induction therapy? There is some mention in the discussion about differences in chemotherapy exposure in obese vs normal weight individuals. But, additional data here would make this report that much more powerful.
Given that obesity could be causally related to some of the factors in the multivariate model, such as LDH or eGFR, I think these variables should be tested for interaction.
Line 166: Data not shown for associations between outcomes in Black females. Why can’t the data be included in a supplemental table?
Other large global studies have identified an association between high BMI and risk of death from myeloma. See for instance: “The global burden of cancer attributable to risk factors, 2010–19: a systematic analysis for the Global Burden of Disease Study 2019” Lancet. 2022.
These studies have also, to my knowledge, not identified a difference associated with sex. One explanation would be that obesity increases the risk of developing MM, but not of more aggressive disease or outcomes once disease has developed. I think that further discussion of this would improve the manuscript.
Minor issues:
-Introduction: Could cite some of the literature on increased risk of MM associated with obesity.
-Discussion line 180-182: I’m not sure it’s that significant to point out that this is the first study to look at obesity at multiple times throughout life given that the only time point that significantly associated with outcome was at the time of diagnosis.
Author Response
We appreciate the thoughtful comments.
Wang et al describe a cohort of patients with MM treated at their institution and look at the association between obesity as measured by BMI and survival in patients with MM. They find no overall association between obesity at any time and MM survival. But, when looking at women and men separately they find an association between obesity and improved outcomes in women only. This is a fascinating finding, and if confirmed will require additional research on the biological underpinnings. The authors should be commended on their clear and well written paper that I think will be of interest to myeloma community at large. I have a few comments/questions that I believe would help improve the quality of the manuscript.
- The biggest question for me is why there might be these differences in survival associated with obesity. Is this due to myeloma related outcomes? One way to look at this is whether lower BMI is associated with any adverse prognostic factors (HR cytogenetics, ISS stage, LDH, lower GFR). Does the normal BMI group include women who are borderline underweight, potentially with aggressive disease mediating weight loss that predisposes the referent population to worse survival?
Response: We thank the reviewer for this important contribution. Please see our responses to Reviewer 1, comment #1.
- Related to this, why is that overweight females live longer? Do they have better myeloma outcomes? Do they have better PFS1 or better ORR to induction therapy? There is some mention in the discussion about differences in chemotherapy exposure in obese vs normal weight individuals. But, additional data here would make this report that much more powerful.
Response: We appreciated the reviewer’s valuable insights. A comprehensive analysis of BMI and/or weight and myeloma outcomes including PFS or therapeutic responses (e.g., complete responses, partial responses, minimal residual disease, etc.) according to the International Multiple Myeloma Working Groups could be informative and an interesting topic for future directions but it is beyond the scope of the current paper. We added a sentence in the limitation (line 301-302) to read “Differences in therapeutic exposure in obese versus normal weight individuals might confound the observed association”
- Given that obesity could be causally related to some of the factors in the multivariate model, such as LDH or eGFR, I think these variables should be tested for interaction.
Response: As suggested by the reviewer, we tested the interaction terms of BMI at diagnosis with age, ISS, LDH, and eGFR in fully adjusted multivariable models. None of the interaction terms were statistically significant. We added this to results section line 175-177.
- Line 166: Data not shown for associations between outcomes in Black females. Why can’t the data be included in a supplemental table?
Response: we have included the Table below as a Supplementary Table 2 in the revision.
Table. Sex- and race- specific associations of body mass index (BMI) at multiple myeloma (MM) diagnosis and overall survival among patients with MM in fully adjusted multivariable modelsa
|
|
White female |
Black female |
White male |
Black male |
|
|
HR (95% CI)a |
|||
|
BMI status |
N=129 |
N= 86 |
N=225 |
N=55 |
|
Normal |
1.0 (referent) |
1.0 (referent) |
1.0 (referent) |
1.0 (referent) |
|
Overweight |
0.97 (0.44, 2.13) |
0.35 (0.15, 0.79) |
1.23 (0.66, 2.28) |
1.42 (0.42, 4.83) |
|
Obese |
0.44 (0.19, 1.05) |
0.19 (0.08, 0.46) |
1.62 (0.85, 3.10) |
4.39 (1.09, 17.70) |
|
HRs: hazard ratios; CI: confidence interval; BMI: body mass index; Q1-Q4: the 1st to the 4th quartiles. a. HRs are based on models that include age (<60, 60-69, ≥70), sex (male/female), race (white/black), international staging system (1/2/3), elevated lactate dehydrogenase levels (normal/elevated), and estimated glomerular filtration rate (>=60/<60). |
||||
- Other large global studies have identified an association between high BMI and risk of death from myeloma. See for instance: “The global burden of cancer attributable to risk factors, 2010–19: a systematic analysis for the Global Burden of Disease Study 2019” Lancet. 2022.
These studies have also, to my knowledge, not identified a difference associated with sex. One explanation would be that obesity increases the risk of developing MM, but not of more aggressive disease or outcomes once disease has developed. I think that further discussion of this would improve the manuscript.
Response: While other studies, including the study by Tran KB, et al., (Lancet 2022), reported an association between high BMI and risk of death, different study designs/populations and methodologies make direct comparisons difficult. For example, most previous studies were based on cohorts of individuals without myeloma at enrollment to evaluate BMI at baseline and deaths from myeloma, whereas our study is composed of newly diagnosed patients with myeloma to evaluate BMI and overall survival. We agree with the reviewer that it’s possible that obesity increases the risk of developing MM, but not of more aggressive disease or outcomes once disease has developed. This is suggested by the studies that we cited and compared with our study with (e.g., reference: 9-11). We discussed this further in the discussion section (line: 257-259).
- Minor issues:
-Introduction: Could cite some of the literature on increased risk of MM associated with obesity.
Response: We have added literatures to the Introduction on increased risk of MM associated with obesity as suggested by the reviewer.
- Discussion line 180-182: I’m not sure it’s that significant to point out that this is the first study to look at obesity at multiple times throughout life given that the only time point that significantly associated with outcome was at the time of diagnosis.
Response: We have deleted this sentence in the 2nd paragraph of discussion.
Round 2
Reviewer 1 Report
Thank you for addressing my comments.